# Seroprevalence of Dromedary Camel HEV in Domestic and Imported Camels from Saudi Arabia

**DOI:** 10.3390/v12050553

**Published:** 2020-05-18

**Authors:** Sherif Aly El-Kafrawy, Ahmed Mohamed Hassan, Mai Mohamed El-Daly, Ishtiaq Qadri, Ahmed Majdi Tolah, Tagreed Lafi Al-Subhi, Abdulrahman A. Alzahrani, Ghaleb A. Alsaaidi, Nabeela Al-Abdullah, Reham Mohammed Kaki, Tian-Cheng Li, Esam Ibraheem Azhar

**Affiliations:** 1Department of Biological Science, Division of Microbiology, Faculty of science, King Abdulaziz University, PO Box 80216, Jeddah 21859, Saudi Arabia; saelkfrawy@kau.edu.sa (S.A.E.-K.); hmsahmed@kau.edu.sa (A.M.H.); mqadery@kau.edu.sa (I.Q.); 2Special Infectious Agents Unit, King Fahd Medical Research Center, King Abdulaziz University, P.O. Box 80216, Jeddah 21589, Saudi Arabia; meldaly@kau.edu.sa (M.M.E.-D.); atoulah@kau.edu.sa (A.M.T.); tlalsobhe@kau.edu.sa (T.L.A.-S.); Nabeela_sa@hotmail.com (N.A.-A.); rmkaki@kau.edu.sa (R.M.K.); 3Department of Medical Laboratory Technology, Faculty of Applied Medical Sciences, King Abdulaziz University, P.O. Box 80205, Jeddah 21589, Saudi Arabia; 4Directorate of Agriculture, Ministry of Environment Water and Agriculture, Makkah Region, Saudi Arabia; dr-a3z@hotmail.com (A.A.A.); gh.vet@hotmail.com (G.A.A.); 5Department of Public Health, College of Nursing, King Abdulaziz University, Jeddah 21859, Saudi Arabia; 6Department of Infection Control and Environmental Health, King Abdulaziz University Hospital, King Abdulaziz University, Jeddah 21859, Saudi Arabia; 7Department of Infectious Diseases, Internal Medicine, Faculty of Medicine King Abdulaziz University, Jeddah 21859, Saudi Arabia; 8Department of Virology II, National Institute of Infectious Diseases, Gakuen 4-7-1, Musashi-murayama, Tokyo 208-0011, Japan; litc@nih.go.jp

**Keywords:** DcHEV, Saudi Arabia, dromedary camels

## Abstract

Hepatitis E Virus (HEV) imposes a major health concern in areas with very poor sanitation in Africa and Asia. The pathogen is transmitted mainly through ingesting contaminated water or food, coming into contact with affected people, and blood transfusions. Very few reports including old reports are available on the prevalence of HEV in Saudi Arabia in humans and no reports exist on HEV prevalence in camels. Dromedary camel trade and farming are increasing in Saudi Arabia with importation occurring unidirectionally from Africa to Saudi Arabia. DcHEV transmission to humans has been reported in one case from the United Arab Emeritus (UAE). This instigated us to perform this investigation of the seroprevalence of HEV in imported and domestic camels in Saudi Arabia. Serum samples were collected from imported and domestic camels. DcHEV-Abs were detected in collected sera using ELISA. The prevalence of DcHEV in the collected samples was 23.1% with slightly lower prevalence in imported camels than domestic camels (22.4% vs. 25.4%, *p* value = 0.3). Gender was significantly associated with the prevalence of HEV in the collected camels (*p* value = 0.015) where males (31.6%) were more infected than females (13.4%). This study is the first study to investigate the prevalence of HEV in dromedary camels from Saudi Arabia. The high seroprevalence of DcHEV in dromedaries might indicate their role as a zoonotic reservoir for viral infection to humans. Future HEV seroprevalence studies in humans are needed to investigate the role of DcHEV in the Saudi human population.

## 1. Introduction

Hepatitis E Virus (HEV) is the causative agent of Hepatitis E globally and imposes a major health concern in Africa and Asia particularly in areas with very poor sanitation [1]. The pathogen is transmitted through ingesting contaminated water or food, coming into contact with affected people and blood transfusions [2]. Old age [3,4], low socioeconomical status [2,5], and illiteracy [6] are among the risk factors for HEV infection. The disease is asymptomatic from the time of infection, but as it manifests, symptoms appear which might include nausea, abdominal pain, vomiting and jaundice [7]. HEV infections are ubiquitous in developing countries as a cause of epidemic and endemic acute hepatitis, the infection can be chronic in immunocompromised individuals receiving chemotherapy, solid organ transplant patients, and HIV patients [8]. However, the disease is now encountered in developed countries as well through ingestion of infected animal products or close contact with infected animals, especially pigs [9]. Mortality from HEV infection is not high except in pregnant women infected with HEV genotype 1 and 2 [10]. HEV infections cause a number of extra-hepatic manifestations, which include a wide spectrum of neurological syndromes [11,12].

HEV belongs to the genus *Orthohepevirus* and family *Orthohepeviridae* [13] and is further classified under two genera namely *Orthohepevirus*, which includes isolates from mammals and chicken and *Piscihepevirus* which includes Cutthroat trout isolates. *Orthohepevirus* includes species A, B, C, and D, while *Piscihepevirus* includes species A only. *Orthohepevirus* A species includes isolates from humans, pigs, wild boars, rabbits, deer, mongoose, and camels [14,15,16]. HEV is a single-stranded, positive- sense RNA genome that varies from 6.6 to 7.3 kB in length. There are eight known HEV genotypes belonging to a particular serotype [17]. The HEV genotypes 1 and 2 can infect humans only, while genotypes 3 and 4 are responsible for infection in humans and animals [13]. Genotypes 5 and 6 are responsible for infection in wild boar [15]. The HEV-7 and HEV-8 were identified in dromedary (1-humped) and bactrian (2-humped) camels, respectively [15,17]. Phylogenetic studies have shown that HEV sequences were detected in dromedaries in the United Arab Emirates and the isolates are classified as a new *orthohepevirus* genotype, HEV 7 [15,18,19]. The first report of HEV type 7 in humans was detected from a patient undergoing a liver transplant and is linked to consuming camel products [20].

Hepatitis E is endemic in many Middle Eastern countries (Turkey, Yemen, Libya, Oman, Bahrain, Iran, Kuwait, Saudi Arabia, and the United Arab Emeritus) [21], some regions of Southeast Asia (Thailand, Singapore) [22], and South America (Brazil, Argentina, Ecuador, and Uruguay) [23]. Hepatitis E is responsible for more than a quarter of all cases of acute intermittent hepatitis and impending hepatitis, however, jaundice epidemics caused by HEV infection do not occur in such areas [24]. The epidemiology of HEV in Egypt is distinct and different from the rest of the world, the disease occurs at a young age. The HEV that affects the Egyptian population is the HEV-1 genotype, with subtypes that are not found in the Asian population [25,26].

Autochthonous HEV has been reported frequently in the developed world and has been linked to the consumption of pork or wild animal products [27,28,29]. HEV was detected in dromedaries from the Middle East and the virus was named DcHEV [15]. About 1.5% of the adult dromedary fecal samples showed the presence of DcHEV RNA [19]. Comparative genomic and phylogenetic analyses showed that DcHEV represents a previously unrecognized HEV genotype and was designated as HEV-7. Recently, the zoonotic potential of DcHEV was reported in a liver transplant patient from the Middle East, who frequently ate camel meat and drank camel milk [20] indicating the possibility of zoonotic transmission of HEV-7 to humans. A recent study indicated that DcHEV prepared by a reverse genetic system resulted in HEV infection in cynomolgus monkeys, providing new evidence of zoonotic infection by DcHEV [30]. The pathogenicity of DcHEV has been unclear and may be multifactorial [30]. Despite the availability of DcHEV genome sequences in the sequence databases, the antigenicity, pathogenicity, and epidemiology of DcHEV is unclear due to the lack of a cell culture system for the virus [31].

The seroprevalence of DcHEV infection has not been well studied, even in areas where camels are most frequently available for use in transportation and for meat and milk, due to the lack of an accurate method for detecting anti-DcHEV antibodies. A previous study showed that an enzyme-linked immunosorbent assay (ELISA) using virus-like particles of DcHEV (DcHEV-LPs) could detect anti-DcHEV IgG and IgM [31] in camels and was utilized for investigating the prevalence of anti-DcHEV in camels from Ethiopia [32].

Very few reports including old reports are available on the prevalence of HEV in Saudi Arabia in humans [24,33,34,35] and no reports exist on the HEV prevalence in camels. Dromedary camel trade and farming have been increasing in Saudi Arabia over the past three decades and importation is unidirectional from Africa to Saudi Arabia [36]. More than 60% of the world’s dromedary camel population lives in East African countries [37]. This has instigated us to perform this investigation of the seroprevalence of HEV in imported and domestic camels to fill in this gap in knowledge in this important health area.

## 2. Materials and Methods

### 2.1. Samples

Imported camels were collected from Jeddah seaport in the period from September 2016 to May 2018. Serum samples were collected from the camels before offloading from the vessels. The camels included in this study were part of a previous study for investigating MERS-CoV in dromedary camels, a full description of the study can be found in [38] Serum samples were collected from 888 imported camels (719 from Sudan and 169 from Djibouti). This study also included 284 serum samples collected from local camel farms and slaughterhouses between July and August 2018 in Jeddah, Saudi Arabia. The serum samples were transported to the Special Infectious Agents Unit, King Fahd Medical Research Center, King Abdulaziz University and stored at −80 °C until testing.

### 2.2. Methodology

Anti-DcHEV IgG-Abs were detected using ELISA following the procedure of Li et al. [32]. Briefly 96-well ELISA microplates (Immulon 2; Thermo Scientific, Rochester, NY, USA) were coated with purified HEV-LPs (1 μg/mL, 100 μL/well) (kindly provided by Dr Tian-Cheng Li) and incubated at 4 °C overnight. The plates were washed twice with PBS/tween-20 (0.05%) (PBS-T) then blocked with 200 μL of 5% skim milk in PBS-T (blocking buffer) for 1 h at 37 °C followed by washing three times with PBS-T. Camels’ sera (100 μL/well) were added at a dilution of 1:200 in blocking buffer. The plates were incubated at 37 °C for 1 h and then washed 3 times. Then, 100 μL of peroxidase-conjugated goat anti-camel IgG (H + L) (1:1000 dilution) (Alpha Diagnostic International, San Antonio, TX, USA) in blocking buffer was added to each well. The plates were incubated at 37 °C for 1 h and washed 3 times with PBS-T. Then 100 µL of tetramethylbenzidine (TMB) substrate (KPL, Gaithersburg, MD, USA) was added to each well and the plates were incubated at room temperature for 30 min. The colorimetric reaction was stopped with 100 µL of 1N HCL (0.8%). Absorbance was measured spectrophotometrically at 650 nm. The cutoff was calculated based on the 20 known negative samples that were tested in triplicate (mean OD + 4 x SD) and was found to be 0.257. All serum samples were tested in duplicate and samples with an optical density above the cut-off values were considered positive. Samples with border line index values (OD/cutoff = 0.8–1.2) were tested in triplicate on three separate runs and the results of two repeats were taken as the final result if a discrepancy between the three repeats was found.

### 2.3. Ethics Statement

The study was conducted after obtaining the needed permits and approvals from the Directorate of Agriculture, Ministry of Environment, Water and Agriculture, Jeddah, Saudi Arabia.

The study was approved by the Unit of Biomedical Ethics, King Abdulaziz University Hospital, (Approval number 16-121, dated 23 Dec 2016).

### 2.4. Statistical Analysis

χ2 or Fisher’s exact test were used as appropriate to compare data. Two-sided *p* value of <0.05 was used as a predictor for statistical significance. Statistical analysis was performed using SPSS version 21 (IBM corporation, Armonk, NY, USA).

## 3. Results

In this study, samples were collected from camels imported from Sudan (719) which included 655 males and 64 females, and camels imported from Djibouti (169) which were all males. The camels’ ages for imported camels ranged from 1 to 5 years. The study also included camel samples that were collected from domestic camel collected from local camel farms and slaughterhouses in the Jeddah area (284) including 244 males and 40 females, their ages ranged from <1 to 5 years.

The total number of positive cases in the collected samples was 270 resulting in a total prevalence of 23.1%. Imported camels showed a slightly lower prevalence than domestic camels (22.4% vs. 25.4%) with no significant difference (*p* value = 0.3). The age of the camels did not show significant difference in the prevalence of HEV in camels (*p* value = 0.827). Gender was significantly associated with the prevalence of HEV in the collected camels (*p* value = 0.015) where males (31.6%) were more infected than females (13.4%) (likelihood ratio = 6.675). The main characteristics of the IgG positive samples are presented in Table 1.

## 4. Discussion

HEV has been considered the cause of acute hepatitis for a long time and is transmitted mainly through a fecal oral route in low socio-economic areas of the world [39]. The virus has now been reported in developed countries where autochthonous hepatitis E is increasingly recognized [40]. It is reported to be acquired zoonotically mainly through consumption of wild animal products while concerns have arisen regarding the safety of blood and blood products [39]. Mandatory blood donation testing was implemented in Europe, Ireland, the United Kingdom, and the Netherlands, while blood donation screening is in consideration in Germany [41,42]. DcHEV has been proposed as a new genotype of HEV (G7) among Orthohepevirus A species with other HEV strains isolated from human, rabbit or wild boar [18]. A long-term epidemiological study confirmed the detection of partial DcHEV genome in camel serum or fecal samples from the UAE, North and East Africa and Pakistan during the period 1983–2015. The study indicated that DcHEV in dromedary camels has been long-established, diversified, and geographically widespread [19].

In this study, we collected samples from imported camels from Africa and from domestic camels from farms and slaughterhouses from Jeddah, Saudi Arabia. The assay used in this study was established by Li et al. [32] based on using DcHEV-LPs as an antigen and was validated for the detection of DcHEV-Abs in Camels from Ethiopia. The overall seroprevalence of DcHEV in the collected samples was 23.1% showing a prevalence of 25.4% in domestic and 22.4% in imported camels. The prevalence of DcHEV in the imported camels was comparable to earlier reports from the African continent including a study that reported a prevalence of 20% in camels from Ethiopia [32]. Another study from Africa showed a slightly higher prevalence in camels from Sudan, Somalia, and Kenya ranging from 31% to 42% [19]. This difference in prevalence between the two studies could be due to either the different assay used or the sampling time frame difference; Rasche et al. investigated samples in the period from 1983–2015 while the samples from Sudan were collected in 1983. The only significant association in our study was found between gender and the positivity of the DcHEV-IgG antibodies (*p* = 0.015) while there was no statistical difference in prevalence between imported and domestic camels. A recent study [43] showed a similar prevalence in dromedaries where they found a seroprevalence of 86.6%.

The high prevalence of HEV in Saudi Arabia reported by old studies showed a seroprevalence ranging from (7.2% vs. 10.8%) in dialysis patients compared to healthy controls [35] to a prevalence of 16.9% and 18% in blood donors in Jeddah (west) and sickle cell anemia children in Jazan (south) [33,34]; respectively. This study is the first study to investigate the prevalence of HEV in dromedary camels from Saudi Arabia. The high seroprevalence of DcHEV in dromedaries indicates that they might play a role as a zoonotic reservoir for viral infection to humans. In the absence of pig farming and consumption in Saudi Arabia, due to religious considerations, camels might play a role in the transmission of HEV to humans. Future HEV seroprevalence studies in humans are needed to investigate the role of DcHEV in the human population in Saudi Arabia especially due to the consumption of camel meat and products in the country. The potential role of dromedary camels in spreading HEV infection to humans is indicated by the experimental transmission of DcHEV to primates [30], together with the report of a possible transmission of HEV from consuming camel meat and milk to a person in UAE [20]. This shows that DcHEV might play a role in the spread of HEV hepatitis in regions where consumption of camel milk and camel products are common. Further studies are needed to investigate the prevalence of HEV RNA in dromedary populations in Saudi Arabia to elucidate the burden of the disease and the prevalent genotypes of HEV in the camel populations. Screening the human population for HEV will give more insights into the potential role of dromedaries in HEV transmission. Due to the fecal oral route of transmission of HEV it is recommended to investigate the seroprevalence of HEV in other animal species that are in frequent contact with camels such as sheep, goats, and horses.

## Figures and Tables

**Table 1 viruses-12-00553-t001:** Main characteristics of the IgG positive camel sera.

		Prevalence % (n)		
		Total	Imported	Domestic
Gender	Male	94.81 (256)	98 (194)	86.11 (62)
	Female	5.19 (14)	2 (4)	13.89 (10)
Age Group	1–3 yrs	20.37 (55)	1.52 (3)	72.22 (52)
	>3 yrs	79.63 (215)	98.48 (195)	27.78 (20)
Collection Year	2016	33.33 (90)	45.45 (90)	0 (0)
	2017	32.96 (89)	44.95 (89)	0 (0)
	2018	33.70 (91)	9.60 (19)	100 (72)

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
