# Peer review of "Seroprevalence of Dromedary Camel HEV in Domestic and Imported Camels from Saudi Arabia"

_viruses, 2020, doi:10.3390/v12050553_

Round 1

Reviewer 1 Report

This is an interesting finding as the anti-HEV prevalence of dromedary is unknown in Saudi Arabia. I suggest the authors to detect HEV RNA in all samples or maybe the current study is more suitable for a more concise format.

Author Response

This is an interesting finding as the anti-HEV prevalence of dromedary is unknown in Saudi Arabia. I suggest the authors to detect HEV RNA in all samples or maybe the current study is more suitable for a more concise format.

Response:

We thank the reviewer for his kind remark and agree with the comment concerning the need to detect HEV RNA in all samples, but the PCR and sequencing of the RNA positive samples will be the subject of another article that we expect to publish soon in this same special issue.

Reviewer 2 Report

INTRODUCTION- The authors should add a description HEV pathogenicity for camel. They prevalently describe HEV infection  in Humans.

MATERIALS &METHODS-a)The authors must indicate the sensitivity and specificity of serological assay.

  1. b) How were considered the samples with O.D. near cut-off?

1.Since interspecies transmission is also been documented,  the authors should include  a serological study in other animal species ( i.e. sheeps, goats, horses) that can have contact with camel. This is relevant for HEV food  transmission  prevention.

2.By serological test, we cant distinguish the HEV genotype that causes infection, thus, the authors must also conduct experiments to detect HEV RNA to assess HEV genotypes, subtypes and variants circultaing in Saudi Arabia among camels.

Author Response

INTRODUCTION-

The authors should add a description HEV pathogenicity for camel. They prevalently describe HEV infection in Humans.

Response:

We thank the reviewer for the comment the details about HEV infection in human comes from the importance of this finding on human health and the potential of camels as animal reservoirs to HEV infection in Saudi Arabia. But the following statement was added to the introduction section “The pathogenicity of DcHEV has been unclear and may be multifactorial [30]. Despite the availability of DcHEV genome sequences in the sequence databases, the antigenicity, pathogenicity, and epidemiology of DcHEV is unclear due to the lack of a cell culture system for the virus [31].” Page 2, lines 89-92.

MATERIALS &METHODS-

  1. The authors must indicate the sensitivity and specificity of serological assay.

Response:

The specificity was confirmed in earlier studies by Dr Li, using rabbit serum immunized with VLP of Norovirus, HPV, BKV, JCV, human bocavriuses and porcine Bocavirus. The HEV-LPs did not show any cross-reaction with above sera. However, it is difficult to indicate the sensitivity, since there is no standard available.

  1. How were considered the samples with O.D. near cut-off?

Response:

Samples around the cut off were retested three times and the results was determined by taking two results of the three if there was a discrepancy between the three results. This statement was added to the materials and methods “Samples with border line index values (OD/cutoff=0.8-1.2) were tested in triplicates on three separate runs and the results of two repeats were taken as the final result if a discrepancy between the repeats were found”

  1. Since interspecies transmission is also been documented, the authors should include a serological study in other animal species ( i.e. sheeps, goats, horses) that can have contact with camel. This is relevant for HEV food transmission prevention.

Response:

The current study is intended to investigate the seroprevalence of HEV in camels, we do not have samples for other animal species to conduct a similar study. But we added in the recommendation section the following statement “due to the fecal oral route of transmission of HEV it is recommended to investigate the seroprevalence of HEV in other animal species that are in frequent contact with camels like sheep, goats and horses” page5, lines 200-202.

  1. By serological test, we cant distinguish the HEV genotype that causes infection, thus, the authors must also conduct experiments to detect HEV RNA to assess HEV genotypes, subtypes and variants circultaing in Saudi Arabia among camels.

Response:

We agree with the reviewer comment that serological testing does not distinguish HEV genotypes. To answer this question, we are conducting another study to investigate the prevalence of HEV RNA, and genotypes based on phylogenetic analysis. This will be another manuscript that we are expecting to publish soon in this special issue. The following statement was added in the discussion section to address this point “Further studies are needed to investigate the prevalence of HEV RNA in dromedary populations in Saudi Arabia to elucidate the burden of the disease and the prevalent genotypes of HEV in the camel populations. Screening the human population for HEV will give more insights into the potential role of dromedaries in HEV transmission.”

Reviewer 3 Report

Dr El-Kafrawy et al. interested in the anti-HEV IgG seroprevalence of local or imported Dromedary camels in Saudi Arabia. The authors found no difference of prevalence between the 2 groups. Interestingly, gender was associated to a higher prevalence, like in humans. The paper is interesting and concise.

Major concerns:

Would it be possible to test the samples for HEV RNA? That would be great to have an idea of the RNA prevalence in the Dromedary Camel.

Table 1 should be put before the discussion section. The line <= 1 should be removed and the second line could be <1-3yrs. More importantly, I cannot find the results in the table. It looks as if it shows the main features of the positive camels but not the results of the IgG prevalence. The total seroprevalence; 270/1172,with an overall prevalence of 23.1% as mentioned in the text, does not appear in the table. So please modify the table to make it easier to read or modify the text “results are presented in table 1” since the table does not contain those results.  

Minor concerns:

Line 53: solid organ transplant patients can also develop chronic infections.

Line 55: “[…] except in pregnant women” I would add “infected with HEV genotype 1 and 2”.

Line 58: ICTV does not recognize the genus Hepevirus anymore. Please only use Orthohepevirus.

Lines 174-176: Could the authors provide some explanation why the IgG prevalence is higher in mal Dc?  This is very intriguing.

The recent paper from Bassal et al. (Epidemiol Infect. 2019) could be discussed in the discussion section.

Author Response

Dr El-Kafrawy et al. interested in the anti-HEV IgG seroprevalence of local or imported Dromedary camels in Saudi Arabia. The authors found no difference of prevalence between the 2 groups. Interestingly, gender was associated to a higher prevalence, like in humans. The paper is interesting and concise.

Major concerns:

Would it be possible to test the samples for HEV RNA? That would be great to have an idea of the RNA prevalence in the Dromedary Camel.

Response:

We agree with the reviewer comment concerning the need to detect HEV RNA in all samples, but the PCR and sequencing of the RNA positive samples will be the subject of another article that we expect to publish soon in this same special issue.

Table 1 should be put before the discussion section.

Response:

Table 1 was replaced before the discussion section.

The line <= 1 should be removed and the second line could be <1-3yrs.

Response:

Line 1 in the table was removed.

More importantly, I cannot find the results in the table. It looks as if it shows the main features of the positive camels but not the results of the IgG prevalence. The total seroprevalence; 270/1172, with an overall prevalence of 23.1% as mentioned in the text, does not appear in the table. So please modify the table to make it easier to read or modify the text “results are presented in table 1” since the table does not contain those results.

Response:

We agree with the reviewer comments. The text was modified to “The main characteristics of the IgG positive samples are presented in table 1”. Page 4, line 55; and the table legend was changed to “Main characteristics of the IgG positive camel sera”

Minor concerns:

Line 53: solid organ transplant patients can also develop chronic infections.

Response:

The statement was changed to “HEV infections are ubiquitous in developing countries as a cause of epidemic and endemic acute hepatitis, the infection can be chronic in immunocompromised individuals receiving chemotherapy, solid organ transplant patients and HIV patients.”

Line 55: “[…] except in pregnant women” I would add “infected with HEV genotype 1 and 2”.

Response:

The Statement was changed to “Mortality from HEV infection is not high except in pregnant women infected with HEV genotype 1 and 2”

Line 58: ICTV does not recognize the genus Hepevirus anymore. Please only use Orthohepevirus.

Response:

The term Hepevirus was replaced with Orthohepevirus.

Lines 174-176: Could the authors provide some explanation why the IgG prevalence is higher in mal Dc? This is very intriguing.

Response:

This is a very interesting remark, but the data could not provide a logical explanation of this observation. Future studies might be needed to investigate the role of gender in the susceptibility to HEV infection.

The recent paper from Bassal et al. (Epidemiol Infect. 2019) could be discussed in the discussion section.

Response:

The study of Bassal et al was discussed in the discussion section and the following statement was added “A recent study [44] has shown a similar prevalence in dromedaries where they found a seroprevalence of 86.6%.”